# Role of RPA Phosphorylation in the ATR-Dependent G2 Cell Cycle Checkpoint

**DOI:** 10.3390/genes14122205

**Published:** 2023-12-13

**Authors:** Shengqin Liu, Brendan M. Byrne, Thomas N. Byrne, Gregory G. Oakley

**Affiliations:** 1Department of Oral Biology, University of Nebraska Medical Center College of Dentistry, Lincoln, NE 68583, USA; 2Eppley Cancer Center, Omaha, NE 68198, USA

**Keywords:** ATM, ATR, RPA

## Abstract

Cells respond to DNA double-strand breaks by initiating DSB repair and ensuring a cell cycle checkpoint. The primary responder to DSB repair is non-homologous end joining, which is an error-prone repair pathway. However, when DSBs are generated after DNA replication in the G2 phase of the cell cycle, a second DSB repair pathway, homologous recombination, can come into action. Both ATM and ATR are important for DSB-induced DSB repair and checkpoint responses. One method of ATM and ATR working together is through the DNA end resection of DSBs. As a readout and marker of DNA end resection, RPA is phosphorylated at Ser4/Ser8 of the N-terminus of RPA32 in response to DSBs. Here, the significance of RPA32 Ser4/Ser8 phosphorylation in response to DNA damage, specifically in the S phase to G2 phase of the cell cycle, is examined. RPA32 Ser4/Ser8 phosphorylation in G2 synchronized cells is necessary for increases in TopBP1 and Rad9 accumulation on chromatin and full activation of the ATR-dependent G2 checkpoint. In addition, our data suggest that RPA Ser4/Ser8 phosphorylation modulates ATM-dependent KAP-1 phosphorylation and Rad51 chromatin loading in G2 cells. Through the phosphorylation of RPA Ser4/Ser8, ATM acts as a partner with ATR in the G2 phase checkpoint response, regulating key downstream events including Rad9, TopBP1 phosphorylation and KAP-1 phosphorylation/activation via the targeting of RPA32 Ser4/Ser8.

## 1. Introduction

DNA double-strand breaks (DSBs) pose a major threat to genomic integrity. Upon the detection of DSBs, cells trigger a DNA damage response (DDR) that arrests the cell cycle and allows time to repair the damaged DNA [1,2]. Primarily, two distinct pathways mediate DSB repair: non-homologous end joining (NHEJ) and homologous recombination (HR)-directed repair. NHEJ makes the initial repair attempt by directly religating the ends with little DNA end processing. However, if rapid rejoining cannot occur due to lesions or complex chromatin, HR-directed repair ensues [3]. HR-directed repair requires DSB end resection, which is a process that removes bases from 5′ ends to generate a long section of single-stranded DNA (ssDNA). While end resection prepares DSB ends for strand invasion for HR, it also creates a platform for ATR and checkpoint activation. Vital to both end resection and ATR activation, Replication Protein A (RPA) is recruited to the newly formed ssDNA region and forms an RPA–ssDNA complex [3,4,5]. The formation of this complex is pivotal to promoting ATR and checkpoint activation in response to DSBs [6]. ATR recognizes RPA-coated ssDNA through the ATR-interacting protein (ATRIP). The canonical ATR signaling pathway is dependent on the colocalization of the ATR–ATRIP complex with the Rad9–Rad1–Hus1 (9-1-1) complex [7]. The basic signaling pathway involves the 9-1-1 complex mediating the recruitment of DNA Topoisomerase II Binding Protein 1 (TopBP1) and consequent ATR activation [8,9].

As a major ssDNA binding protein in mammalian cells, RPA is essential for numerous DNA metabolic processes [10,11]. In response to perturbations of DNA, RPA is phosphorylated within the first 33 residues of the N-terminus of RPA32 subunit, in which nine sites have been identified [12,13,14]. Each of these unique phosphorylated species allows RPA to function in DNA pathways. For example, the replication machinery is able to distinguish between RPA species with different phosphorylation states, which allows for the regulation of DNA replication as well as the identification of DNA damage or replication stress sites for the recruitment of repair factors [15,16]. In response to DNA damage, RPA32 is phosphorylated at S/TQ and non-S/TQ PIKK consensus sites, Thr21/Ser33 and Ser4/Ser8/Ser12, respectively, by the three major phosphatidylinositol kinase-related kinases (PIKKs), ataxia telangiectasia and rad-3-related kinase (ATR), ataxia telangiectasia mutated kinase (ATM) and DNA-dependent protein kinase catalytic subunit (DNA-PKcs) [11]. Complex priming and reciprocal priming by cyclin-dependent kinase (CDK) and PIKKs occurs, resulting in what is considered the mature hyperphosphorylated species of RPA32 [14,17]. Previous studies have determined that the phosphorylation of RPA32 Ser33 is involved in the progression and repair of stressed replication forks and is exclusively phosphorylated by ATR [18,19]. There is considerable work that has described the kinetics of ATR phosphorylation of Ser33 in response to replication-associated DSBs. For example, in response to replication-dependent DNA damage induced by the topoisomerase I inhibitor, camptothecin, RPA32 can be phosphorylated at Ser33 in two distinct modes [19]. In the first mode, ATR phosphorylates RPA as part of a RPA–ssDNA complex located near ssDNA/dsDNA junctions. In the S phase, ATR phosphorylates RPA32 Ser33 progressively during DSB end resection prior to the phosphorylation of Ser4/Ser8 by DNA-PKcs. In the second mode, the ATR phosphorylation of Ser33 is dependent on a direct interaction between Nbs1 and RPA [19,20]. Zou and colleagues further suggest that during DNA replication, the phosphorylation of Ser33 may be the initial phosphorylation event on RPA32 that is driven by end resection.

While characterized as completing the maturation of the hyperphosphorylated form, the function of RPA32 Ser4/Ser8 phosphorylation has not been as well defined as Ser33. Early research examining RPA32 Ser4/Ser8 phosphorylation demonstrated that this hyperphosphorylated form does not associate with sites of chromosomal DNA replication; however, more recently, it has been used as a readout for efficient DNA end resection [15,21,22].

In this study, we examine the significance of RPA32 Ser4/Ser8 phosphorylation in response to DNA damage, specifically contrasting DNA damage and ATR signaling in the S phase to G2 phase of the cell cycle.

## 2. Materials and Methods

### 2.1. Cell Culture and Treatments

Head and neck squamous cell carcinoma cells, UMSCC-38, were cultured in Dulbecco’s modified Eagle medium (Invitrogen Waltham, MA, USA) supplemented with 10% fetal bovine serum (Hyclone), 100 µg/mL penicillin and 100 µg/mL streptomycin and maintained at 37 °C and 5% CO_2._ The UMSCC-38 cells established at the University of Michigan were generously provided by Dr. Thomas Carey (University of Michigan). The SV40-transformed fibroblast cell line AT22IJE-T contains a homozygous frameshift mutation at codon 762 of the *ATM* gene. The cell lines AT22IJE-pEBS7 and AT22IJE-YZ5, generous gifts from Y. Shiloh (Tel Aviv University, Ramat Aviv, Israel), were derived from the SV40- transformed fibroblast line AT22IJE-T by transfection with either the empty episomal expression vector pEBS7 or with the ATM cDNA clone pEBS7-YZ5, allowing the constitutive expression of the functional recombinant ATM protein [23,24]. The SV40-transformed fibroblasts were grown in Dulbecco’s modified Eagle’s medium (GIBCO-BRL, Waltham, MA, USA) supplemented with 15% fetal bovine serum (GIBCO-BRL, Waltham, MA, USA) and 0.1 mg/mL hygromycin (Calbiochem, La Jolla, CA, USA).

### 2.2. Synchronization of Cells and Treatment with Etoposide

Cells were synchronized using a double thymidine block. To synchronize cells in the S phase, cells were grown to subconfluent density and then treated with 2.0 mM thymidine for 18 h, washed thoroughly with PBS, and released into fresh thymidine-free medium for 9 h, which was followed by the addition of thymidine again for 18 h. Thymidine was removed for a second time, and cells were released for 4 h and verified in the S phase by FACS analysis (Figure 1A). To synchronize in the G2 phase, the same steps were followed as for the S phase synchronization except that the cells were released from the second thymidine block for 8 h. We then verified the synchronization in the G2 phase by FACS analysis. Once in the S or G2 phase, cells were incubated in growth medium containing 20 µM etoposide for 2 h (indicated on the timeline as “−2” to “0”); then, etoposide was removed and replaced with fresh medium and incubated for indicated times and harvested.

### 2.3. Cell Lysates and Chromatin Isolation

To prepare cell lysates, cultured cells were harvested by centrifugation, washed in phosphate buffer saline (PBS), and resuspended for 10 min on ice in cell lysis buffer (50 mM Tris-HCl, [pH 7.5], 150 mM NaCl, 0.1% Nonidet P-40, 10 mM NaF, 10 mM b-glycerophosphate, 1 mM Na_3_VO_4_, and protease inhibitor cocktail, Calbiochem, Burlington, MA, USA) containing 0.5% Triton X-100. Briefly, to isolate chromatin, the cells were washed once with PBS and incubated on ice in buffer A (10 mM HEPES, (pH 7.9), 10 mM KCl, 1.5 mM MgCl_2_, 0.34 M sucrose, 10% glycerol, 1 mM DTT, 10 mM NaF, 10 mM b-glycerophosphate, 1 mM Na_3_VO_4_, and phosphatase inhibitor cocktail) with 0.1% Triton X-100 for 5 min. Nuclei were collected by centrifugation (10 min, 1300× *g*, 4 °C) and washed once with buffer A and lysed in buffer B (3 mM EDTA, 0.2 mM EGTA, 1 mM DTT, 10 mM NaF, 10 mM b-glycerophosphate, 1 mM Na_3_VO_4_, and phosphatase inhibitor cocktail) for 30 min on ice. Insoluble chromatin was collected by centrifugation (5 min, 1700× *g*, 4 °C). The final chromatin pellet was resuspended in SDS sample buffer, boiled and separated by SDS-PAGE.

### 2.4. Cloning and Creation of Transgenic Cell Lines

For the replacement of endogenous RPA32, wild-type and the S4A/S8A mutant retroviral vectors were generated. A fragment representing amino acids 18–271 was amplified from an RPA32-containing pET-11d vector (generously supplied by Marc Wold, University of Iowa) using primers 5′-GCGCACCGGTGATATACAT ATGTGGAAC-3′ and 5′-CGCGGGATCCGTAAGCTCAGTAATCTGGAACATCGTATGGGTATTCTGCATCTGTGGA-3′ and digested with BamHI and NaeI, effectively placing an HA-tag on the 3′ end. Double-stranded oligonucleotides, pertaining to amino acids 1-18 of wild-type and alanine-substituted Ser4/Ser8 RPA, were synthesized with AgeI and NaeI overhangs. The gene fragment and oligonucleotides were ligated into AgeI/BamHI digested pQCXIH, creating a wild-type construct (wt) and a construct with substituted alanines at Ser4 and Ser8 (S4A/S8A). To create recombinant retrovirus, Phoenix cells (Orbigen, San Diego, CA, USA) were plated in 60 mm culture dishes at 5 × 10^6^ cells/mL and transfected with 24 µg of wild-type or S4A/S8A RPA with Lipofectamine 2000 (Invitrogen). After 48 h at 37 °C, cells were incubated for an additional 24 h at 32 °C. Supernatants were collected, centrifuged (10 min at 2000× *g*) and added to 25% confluent UMSCC38 cells in the presence of 10 µg/mL polybrene (Sigma-Aldrich, St. Louis, MO, USA). After 48 h, cells were selected with 20 µg/mL hygromycin (Sigma). After selection, the wild-type and S4A/S8A containing UMSCC38 cells were infected again, this time with supernatant from Phoenix cells transfected with a retroviral shRNA vector targeting the 3′ untranslated region of endogenous RPA32 (kindly supplied by Dr. Wu at the Scripps Institute) and selected with 150 µg/mL G418.

### 2.5. Immunoprecipitation and Immunoblotting

Cell lysates were separated by SDS-PAGE and transferred to polyvinylidene difluoride (PVDF) membranes. Membranes were probed with primary antibodies to RPA32, p-S4/S8-RPA2, p-S33-RPA, TopBP1, Rad9, ATR (Bethyl Laboratories, Montgomery, TX, USA), RPA32 (Thermo Fisher Scientific, Waltham, MA, USA, Lab Vision, San Francisco, CA, USA), Chk1 (G-4), (Santa Cruz Biotechnology, Dallas, TX, USA), Chk2 (Sigma), p-S345-Chk1, p-T68Chk2, (Cell Signaling Technology, Danvers, MA, USA), H2AX (New England Biolabs, Ipswich, MA, USA) and Orc2 (BD Biosciences) followed by Alexa-Fluor 680-conjugated anti-rabbit (Invitrogen) and DyLight 800-conjugated anti-mouse (ThermoFisher Scientific). To immunoprecipitate RPA32 immune complexes from wt-and S4A/S8A-RPA32 expressing cells, cells were washed with PBS, resuspended in cell lysis buffer for 30 min on ice, and centrifuged for 20 min at 20,000× *g*. The pre-cleared supernatants were incubated with PS4/S8 or HA antibodies at 4 °C overnight followed by incubation with protein A/G agarose (Thermo Fisher Scientific, Pierce, Waltham, MA, USA) for 1 h at room temperature. Proteins from the immunoprecipitations were separated by SDS-PAGE, blotted onto PVDF membranes, and probed with primary antibodies to RPA32 (Thermo Fisher Scientific, Lab Vision), RPA70 and TopBP1 (Bethyl Laboratories) followed by Alexa-Fluor 680-conjugated anti-rabbit (Invitrogen) and DyLight 800-conjugated anti-mouse (Thermo Fisher Scientific) secondary antibodies. To test the recruitment of TopBP1 directly to RPA bound to ssDNA, initially ~500 pmol of a biotinylated oligo-deoxythymidine 30 nt in length was added with 40 µL of streptavidin-coated magnetic beads (Invitrogen). Unbound biotinylated ssDNA was washed away, and ~300 pmol of purified RPA or SSB (Sigma) was added. After unbound proteins were removed, ~200 pmol purified TopBP1 or GST-Rad9 was added. All incubations were completed at room temperature. The proteins bound to biotinylated ssDNA were eluted, separated by SDS-PAGE and detected by immunoblotting with primary antibodies to TopBP1, RPA70 (Bethyl Laboratories), Rad9 (BD Biosciences, San Jose, CA, USA) and RPA32 (Thermo Fisher Scientific Waltham, MA, USA, Lab Vision, San Francisco, CA, USA). Recombinant RPA and Asp-RPA were purified according to Henricksen et al., GST-Rad9 according to Sun et al. and TopBP1 according to Choi et al. All images were obtained with an Odyssey Imager (LI-COR Biosciences, Lincoln, NE, USA).

### 2.6. Flow Cytometry

Cell cycle progression was monitored in cells before and after etoposide treatment, incubation for indicated times and fixation in 70% ethanol overnight. Cells were permeabilized (0.25% Triton X-100 on ice for 15 min), washed, incubated overnight in PBS containing 0.1% BSA, washed and incubated in goat anti-mouse Alexa Fluor 647 antibody for 30 min at room temperature. Cells were incubated in 50 µg/mL propidium iodide and 100 µg/mL RNase A for 30 min. Cells (10,000/sample) were analyzed on a BD FACSarray (BD Biosciences) using 532 and 635 nm excitations and collecting fluorescent emissions with filters at 585/42 nm and 661/16 nm (yellow and red parameters, respectively). BD FACSarray and WinList™ (Verity House, Topsham, ME, USA) software were used for data collection and analysis, respectively.

### 2.7. Immunofluorescence Microscopy

Cells were grown on 22 mm coverslips overnight prior to etoposide treatment. Cell synchronized in G2, non-treated and treated with etoposide as described and fixed at 4 h. After an initial wash with PBS, cells were extracted with PBS containing 0.5% Triton X-100 for 2 min on ice and fixed with 4% paraformaldehyde for 15 min. Next, the coverslips were blocked with 15% goat serum at room temperature and then incubated with primary antibodies to HA (Covance, Princeton, NJ, USA), TopBP1, PS33-RPA32 (Bethyl Laboratories) or CENP-F (Santa Cruz) in blocking solution for 1 h. The coverslips were washed with PBS and incubated with an appropriate Alexa-Fluor 488- or Alexa-Fluor 568-conjugated antibody in blocking solution for 1 h. Cells were mounted in PermaFluor (Fisher) supplemented with 0.5 µg/mL DAPI (Roche, Basel, Switzerland). Immunofluorescent images were captured digitally with a Zeiss Axiovert 200M microscope. Mathematica (Wolfram Research, Champaign, IL, USA) was used to crop individual nuclei from the original images (at 1.6 × 1.6 the average nuclei width) based on the center point of their DAPI staining. Overlapped, blurred, or high background stained nuclei were excluded from the analyses. The region of interest for foci selection was defined as the red (HA or CENP-F)/green (TopBP1 or PS33-RPA32) channel staining within the borders of the DAPI stain. The mean background intensity for each nucleus’s red and green channel was subtracted from the image’s channels and was calculated as the mean intensity outside the region of interest. Individual foci were isolated within the region of interest by finding local maxima points after thresholding each channel (same threshold for all the nuclei samples for each channel) to give the best foci definition; the threshold was chosen for each channel by plotting the mean foci count vs. change in threshold and selecting the threshold giving the highest mean value collectively. The lowest foci count grouping was determined from the mean foci for both channels of the cell line’s controls. Higher foci count groupings were set in equal increments that gave the best sensitivity between the increments. Foci count categories were kept equal for each protein. In total, 200 nuclei were counted per sample. Standard deviation was calculated from three replicate experiments.

## 3. Results

### 3.1. RPA Phosphorylation Affects ATR Signaling in G2 Synchronized Cells

The phosphorylation of Chk1 and RPA32 Ser33 are well established as indicators of ATR activity [14,19,25,26,27,28]. Previous work has demonstrated that RPA phosphorylation plays a role in checkpoint activation [14,27,29,30,31,32]. However, the exact mechanisms have not yet been determined. In order to assess the relationship between RPA phosphorylation and ATR signaling, we replaced endogenous RPA32 in human UMSCC-38 cells by siRNA knockdown and the expression of siRNA-resistant HA-tagged WT or S4A/S8A RPA32. Cells were synchronized in the S and G2 phases and validated by FACS analysis (Figure 1A). Cells treated with the TopII inhibitor, etoposide, were analyzed for activation of the ATR substrates Chk1 and RPA32 Ser33. As expected, Chk1 activity increased in WT RPA32 expressing cells synchronized in the S phase (Figure 1B). Chk1 phosphorylation occurred during etoposide exposure, continued after etoposide removal, and later decreased at 10 h. S4A/S8A RPA32-expressing cells activated and sustained ATR-dependent Chk1 phosphorylation, demonstrating that Ser4/Ser8 phosphorylation of RPA32 had a minimal effect on Chk1 phosphorylation (1B, top panel). In contrast, replacing Ser4/Ser8 with alanines did have an initial effect on phosphorylation of Ser33 (1B, lower panel). During the 2 h etoposide exposure, Ser33 phosphorylation was initially delayed. However, after etoposide removal, there was robust phosphorylation in WT and phospho-mutant-expressing cells of Ser33. These results suggest that the phosphorylation status of RPA32 Ser4/Ser8 has little to no effect on ATR activity in the S phase. This is further validated looking at cell cycle progression. With or without RPA Ser4/Ser8 phosphorylation, cells in the S phase exhibited slowed progression through the S phase and likewise accumulated and activated a G2 arrest (Figure 1C).

In G2 phase synchronized cells, Chk1 phosphorylation continued after etoposide removal in cells with RPA phosphorylation. In contrast, a loss of RPA32 Ser4/Ser8 phosphorylation resulted in decreased Chk1 phosphorylation (Figure 1D, top panel). In agreement with decreased Chk1 phosphorylation, a loss of RPA32 Ser4/Ser8 phosphorylation resulted in the decreased phosphorylation of Ser33 (Figure 1D, lower panel). DNA damage-dependent RPA phosphorylation led to a sustained G2 arrest in cells synchronized in G2 (Figure 1E). In contrast, there was an increase in the G1 population and a loss of G2 arrest with the phospho-mutant RPA32 cells synchronized in G2 (Figure 1E). Overall, these findings contrast the differences in the S and G2 phases of the cell cycle with respect to the RPA32 Ser4/Ser8 phosphorylation effect on ATR signaling.

To verify that the replication-independent G2 checkpoint was dependent on Chk1 activity, we inhibited Chk1 phosphorylation with SB218078, which is a selective indolocarbazole-based structure Chk1 inhibitor. Chk1 inhibition led to an almost complete loss of the G2 checkpoint in cells expressing WT RPA32 (Appendix A), demonstrating that ATR and Chk1 are required for G2 replication-independent G2 arrest following DNA damage as previously shown [33].

### 3.2. Initial Chk1 Phosphorylation during DNA Damage in G2 Occurs Independently of RPA Phosphorylation

Next, we asked whether RPA32 Ser4/Ser8 phosphorylation promotes the initial phosphorylation of Chk1 in G2. In the S phase, RPA phosphorylation is a late event in response to replication-dependent DNA damage induced by camptothecin relative to induction of Chk1 activation [19]. In G2, to test whether the initial Chk1 phosphorylation was RPA phosphorylation-dependent, we analyzed Chk1 and RPA32 Ser4/Ser8 phosphorylation at earlier time points (Figure 2). Chk1 phosphorylation was detectable 1 h after etoposide exposure in WT RPA32 and S4A/S8A RPA32-expressing cells (Figure 2, upper panel, −1 h time point). This result would suggest that RPA32 Ser4/Ser8 phosphorylation is not required for the initial activation of ATR phosphorylation of Chk1. However, these data suggest a temporal link between the intensity of Chk1 phosphorylation and RPA32 phosphorylation. Upon initial drug removal, amplified Chk1 phosphorylation was detected and maintained for at least 4 h after drug removal (Figure 2, upper panel), which was not observed in phospho-mutant cells. This increase in Chk1 phosphorylation coincided with an increase in RPA32 phosphorylation in the WT cells (Figure 2, lower panel). This would suggest that RPA32 phosphorylation stimulates and sustains Chk1 phosphorylation but is not required for initial Chk1 phosphorylation in G2.

### 3.3. RPA Phosphorylation Promotes Chromatin Retention and Accumulation of TopBP1 and Rad9 in G2

The RPA–ssDNA complex is required for the initiation of canonical ATR-dependent checkpoint signaling [16]. Our findings indicate that RPA phosphorylation is required for ATR-sustained phosphorylation of Chk1 in G2. We speculated that RPA phosphorylation might be required to stimulate the recruitment and retention of RPA itself or the protein–DNA assembly required to activate the ATR–ATRIP kinase complex. To test this hypothesis, we examined the impact of RPA phosphorylation on chromatin loading of the proteins involved in ATR activation during S and G2 phase exposure to etoposide. During the S phase, RPA displayed increased recruitment to chromatin in WT and phospho-mutant expressing cells, which correlated with the increase in chromatin-bound TopBP1 and Rad9 (Figure 3A). The levels of the replication factor ORC2 are shown as a loading control because the association of ORC2 with chromatin does not change in response to DNA damage [34]. The lack of an impact on the accumulation and retention of TopBP1 and Rad9 to damaged chromatin indicates that RPA32 Ser4/Ser8 phosphorylation does not play a role in protein recruitment required for ATR signaling in the S phase.

In G2-synchronized cells, WT and S4A/S8A RPA32 expressing cells differed in their accumulation of TopBP1 and Rad9 on the chromatin. TopBP1 and Rad9 bound to chromatin increased with RPA phosphorylation, while S4A/S8A RPA32-expressing cells did not support an increased accumulation of TopBP1 and Rad9 (Figure 3A, bottom panel). Interestingly, the binding of RPA to chromatin did not differ between WT and phospho-mutant cells. To further validate the correlation of RPA32 phosphorylation with a chromatin accumulation of TopBP1, we compared the foci formation of RPA32 and TopBP1 at 4 h after etoposide removal. In line with chromatin retention, there was a marked defect in TopBP1 foci formation in the absence of RPA S4/S8 phosphorylation after etoposide treatment (Figure 3B). This was confirmed through counting, indicating that without RPA phosphorylation, there were significantly fewer TopBP1 foci in cells (Figure 3C).

### 3.4. RPA32 Ser33 Phosphorylation Indirectly Requires ATM Activity

RPA32 is progressively phosphorylated by ATR at Ser33 in response to replication-associated DSBs during the S phase [19]. This gradual increase in the phosphorylation of RPA32 Ser33 has been suggested to be a resection-dependent phosphorylation event on RPA32. In G2, DNA end resection and the consequential ATR kinase activity are dependent on the ATM response to DSBs [35,36]. To examine the effect that ATM expression has on Ser33 phosphorylation, we used G2 phase-synchronized ATM-deficient AT cells and the identical AT cells with reintroduced ATM expression (Figure 4). The abundant RPA32 Ser33 foci in AT cells with reintroduced ATM expression and the lack of Ser33 phosphorylation in AT cells indicate that RPA32 Ser33 indirectly requires ATM activity.

### 3.5. RPA32 Ser4/Ser8 Phosphorylation Is Required for KAP-1 Phosphorylation

ATM initiates the steps of DSB end resection by stimulating the nucleolytic activities of CtIP and Mre11, which generates an ssDNA platform for an RPA nucleoprotein filament recognized by the proteins necessary for ATR signaling. The loss of RPA32 Ser4/Ser8 phosphorylation blocks the ATR phosphorylation of Chk1 and the G2 checkpoint. To investigate whether RPA32 Ser4/Ser8 phosphorylation has an effect on ATM activity, we looked at the ATM-dependent phosphorylation of KAP-1 Ser824, which is a substrate of ATM required for DSB repair within chromatin. Following etoposide treatment, KAP-1 Ser824 phosphorylation was drastically reduced in the S4A/S8A mutants up to 10 h after treatment removal (Figure 5A). Our previous study demonstrated that under the same experimental conditions in S phase cells, KAP-1 Ser824 phosphorylation was not affected [14]. To ensure that this is not a unique phenomenon to the topoisomerase inhibitor, etoposide, we examined KAP-1 Ser824 after treatment with irradiation (IR). Consistent with etoposide-treated cells, the S4A/S8A RPA32 mutant displayed substantially decreased KAP-1 phosphorylation levels in response to IR irradiation (Figure 5B).

### 3.6. RPA32 Phosphorylation Influences Phosphorylation of H2AX and Rad51 Chromatin Loading

Phosphorylated H2AX (γH2AX) flanks DSBs and helps provide a signal for the recruitment of DNA damage machinery. The accumulation of γH2AX and other repair proteins assists in the formation of detectable foci that disperse as DNA damage is repaired, making γH2AX a reliable marker of DNA damage and DNA repair over time. With and without RPA32 phosphorylation, cells showed a marked increase in γH2AX during and after etoposide removal. While γH2AX showed a normal profile and began to decrease in WT cells, γH2AX persisted in the absence of RPA32 phosphorylation at 10 h (Figure 5C).

Rad51 recruited to chromatin is closely related to increased HR activity [37,38]. This led us to investigate the recruitment of Rad51 following etoposide exposure. We observed that at later time points (Figure 4C, 4 h and 10 h), there was a slight increase in Rad51 chromatin loading with RPA32 phosphorylation (Figure 5C). Combined, these data suggest a direct or indirect role for RPA32 Ser4/Ser8 phosphorylation in the process of DNA repair.

## 4. Discussion

Here, we have demonstrated that RPA32 Ser4/Ser8 phosphorylation was not required for an S phase ATR-initiated checkpoint; however, in G2 synchronized cells, RPA32 Ser4/Ser8 phosphorylation is necessary for increases in TopBP1 and Rad9 accumulation on chromatin and full activation of the ATR-dependent G2 checkpoint. In addition, our data suggest RPA Ser4/Ser8 phosphorylation modulates ATM-dependent KAP-1 phosphorylation and Rad51 chromatin loading.

Recent work by Finkelstein and colleagues has built on the idea of RPA32 Ser4/Ser8 as a readout of DNA end resection by showing that the hyperphosphorylated form (RPA32 Ser4/Ser8) of RPA inhibits DNA end resection. Using a single molecule technique, they demonstrated that BLM together with RPA stimulated resection by EXO1 and DNA Replication Helicase/Nuclease 2 (DNA2) nucleases. However, as the hyperphosphorylated form replaced unphosphorylated RPA on single-stranded DNA, DNA end resection drastically slowed down. The mechanism proposed is that the mature hyperphosphorylated form of RPA inhibits the RPA1 interaction with BLM, which has been previously shown with other RPA–protein interactions [39]. Extrapolating on Finkelstein’s work, the decrease in efficient DNA end processing in the Ser4/Ser8 mutant cells may alter the recruitment of the 9-1-1 complex and TopBP1, leading to decreased ATR activation. The loss of sustained and amplified Chk1 phosphorylation combined with the G2 checkpoint defect supports the hypothesis that the phosphorylation of Ser4/Ser8 is required for maintaining the proper functioning of the G2 checkpoint. We provide further evidence to support the hypothesis that RPA Ser4/Ser8 phosphorylation is required to amplify and sustain ATR activity. The very early detection of Chk1 phosphorylation occurs independently of RPA32 Ser4/Ser8 phosphorylation in G2 cells; however; the increase in RPA32 Ser4/Ser8 phosphorylation corresponds with an increase in Chk1 phosphorylation that does not occur with the absence of Ser4/Ser8 phosphorylation. This combined with the loss of DSB-induced G2 arrest in cells expressing S4A/S8A RPA32 suggests Ser4/Ser8 phosphorylation is an important event that is correlated with increased and sustained ATR kinase activity.

In the S phase, RPA32 Ser33 phosphorylation is considered the first RPA phosphorylation event in response to replication stress. Not only is RPA32 Ser33 phosphorylation a primary RPA phosphorylation event, it is also important for recovery from replication stress [18,40]. While the phosphorylation of both Chk1 and RPA32 Ser33 occurs through ATR activity, they do so with distinct kinetics and through overlapping yet distinct signaling pathways in the S phase. Chk1 Ser345 requires the Rad17–RFC complex, and RPA32 Ser33 phosphorylation relies heavily on the interaction between RPA and Nbs1, which is part of the MRN complex [19,20]. This may explain in part why RPA32 Ser4/Ser8 phosphorylation in response to UV, hydroxyurea (HU) or IR does not play a role in initial Chk1 activity or sustained Chk1 phosphorylation in S phase cells [27]. Consistent with these findings, we found that Chk1 phosphorylation did not require RPA32 Ser4/Ser8 phosphorylation in S phase cells in response to etoposide-induced DNA damage. Furthermore, in S phase cells, the G2 checkpoint is intact in cells expressing the RPA phospho-mutant, suggesting a redundant mechanisms for ATR signaling.

The effects of S4/S8 RPA32 phosphorylation on ATM are not entirely clear. In response to DNA damage in cells synchronized in G2, ATM is responsible for KAP-1 phosphorylation during early transient events and at later residual phosphorylation time points [41,42,43]. The early and residual KAP-1 phosphorylation occurred normally in cells expressing WT RPA. Cells expressing S4A/S8A RPA32 had substantially decreased early KAP-1 phosphorylation and no detectable residual KAP-1 phosphorylation [43]. Cells expressing a KAP-1 Ser824A mutant induce a DSB defect even when ATM is active [41]. Interestingly, the retention of yH2AX foci with the absence of RPA32 S4/S8 phosphorylation suggests a phenotype similar to cells unable to phosphorylate KAP-1. It is well documented that ATM is necessary to activate ATR in response to DSBs in G2 [44]. AT cells do not phosphorylate Ser4/Ser8 [14], and also Ser33 as shown here in response to G2 DNA damage, indicating a lack of ATR activity toward the RPA substrate. Following this model, this would indicate that Ser4/Ser8 phosphorylation is an important event for further phosphorylation events.

Previously, cells expressing phosphomimetic RPA32 mutants impeded homologous recombination (HR) via an inefficient loading of the essential HR factor Rad51 [40]. Paradoxically, here, we observed a decrease in Rad51 loading with cells expressing phospho-mutant RPA32. One possibility is that the differences in the timing of phosphorylation and dephosphorylation are important in refining Rad51 loading. How RPA alters Rad51 loading is not entirely understood. The phosphorylation of RPA is a transient event that combined with the dephosphorylation and ubiquitylation of RPA are important post-translational modifications that contribute to the fine tuning of the DNA damage response. In support of this, the phosphorylation, dephosphorylation and ubiquitylation of RPA are required for an efficient DNA damage response, facilitating homologous recombination-directed repair and the recovery of stalled replication forks [40,45,46,47].

In conclusion, we showed in the present study the essential role of RPA in regulating the cell response to DNA damage in G2. Targeting RPA phosphorylation could lead to genomic instability through inhibition of the DNA damage response that may in turn provide opportunities for therapeutic strategies.

## Figures and Tables

**Figure 1 genes-14-02205-f001:**
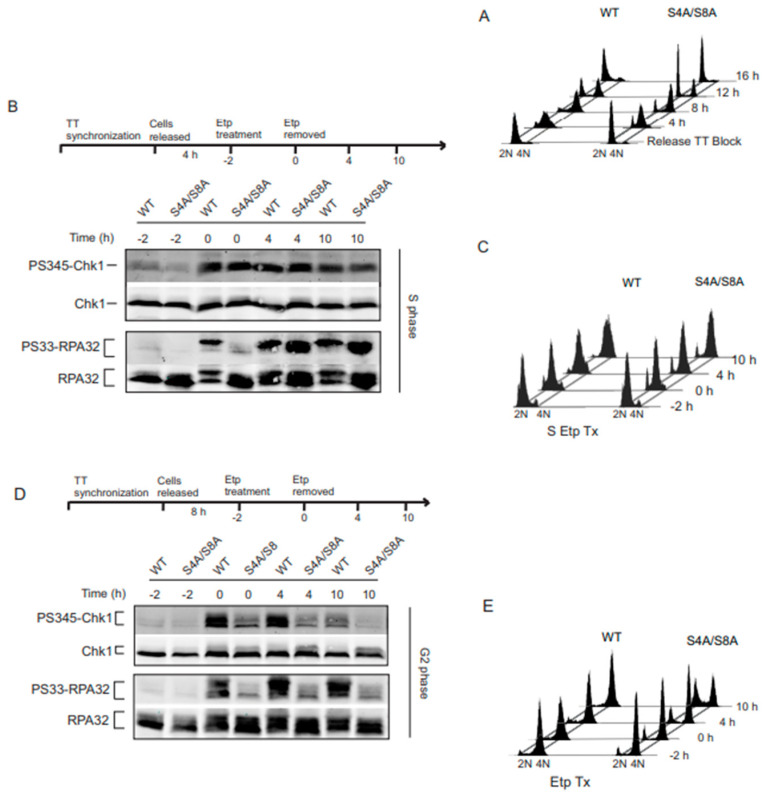
(**A**) FACS analysis of cells following synchronization. (**B**,**D**) Cells expressing HA-tagged WT or S4A/S8A RPA32 with endogenous RPA32 down-regulated were synchronized in S or G2 phase and treated with etoposide (20 µM) from −2 to 0 h. The proteins were resolved by SDS-PAGE followed by immunoblotting with antibodies to the indicated proteins. (**C**,**E**) Cell cycle profiles represent the indicated times the cell lysates were used for protein expression analysis.

**Figure 2 genes-14-02205-f002:**
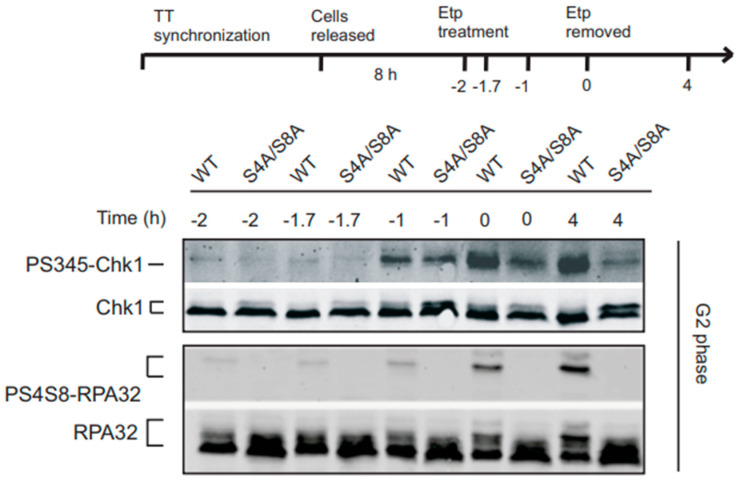
Cells synchronized in G2 phase were treated with etoposide for 2 h followed by replacement with fresh media. Chk1 (Ser345) and RPA32 (Ser4/Ser8) phosphorylation were analyzed by Western blot during and after etoposide removal.

**Figure 3 genes-14-02205-f003:**
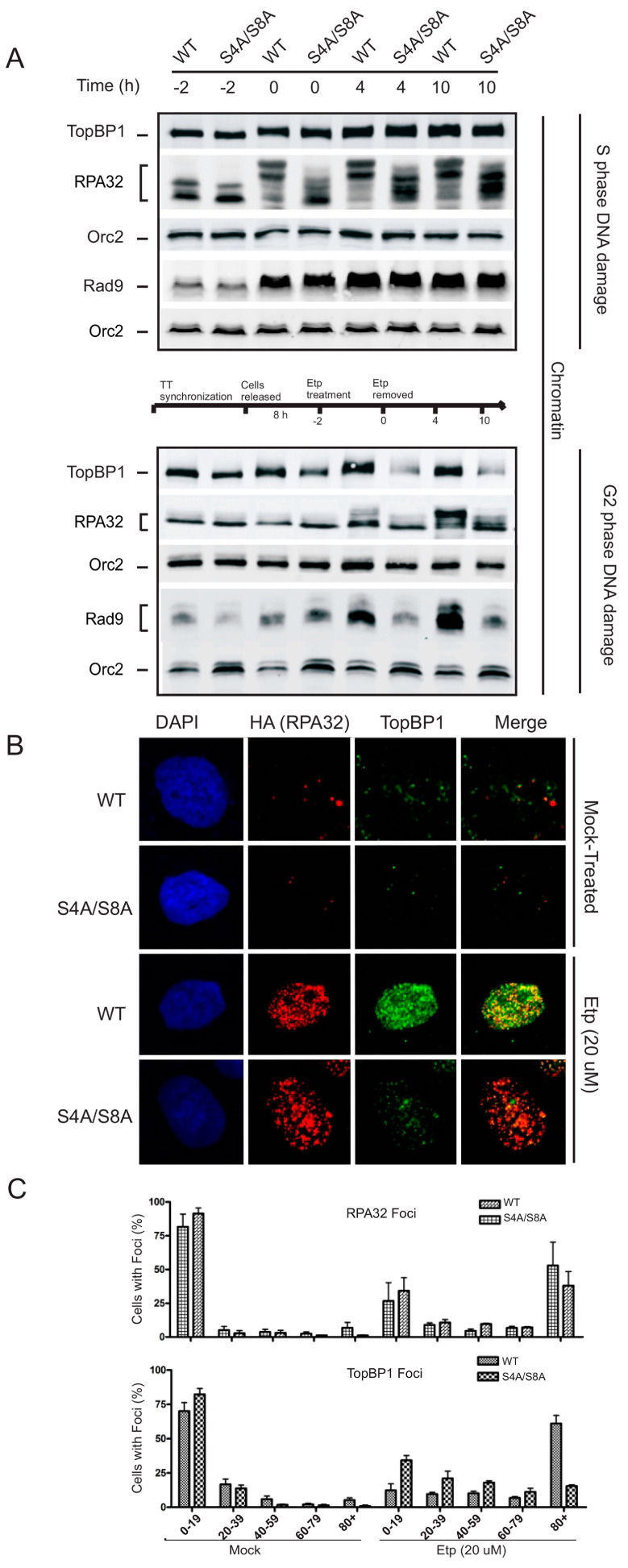
(**A**) WT and S4A/S8A-expressing cells treated with etoposide (20 µM) for 2 h (-2 indicates start of etoposide treatment) followed by drug removal (0 h) for the indicated times were assessed for chromatin retention in S phase (upper panel) and G2 phase cells (lower panel). Orc2 was used as a loading control in each case. (**B**) Immunofluorescence of cells that were detergent washed before fixation. Foci formation images comparing the chromatin accumulation of TopBP1 and RPA32 for etoposide (20 µM) and mock-treated cells at 4 h after etoposide removal. (**C**) Calculated percentage of cells containing RPA and TopBP1 foci in a total of 200 cells per condition for each experiment.

**Figure 4 genes-14-02205-f004:**
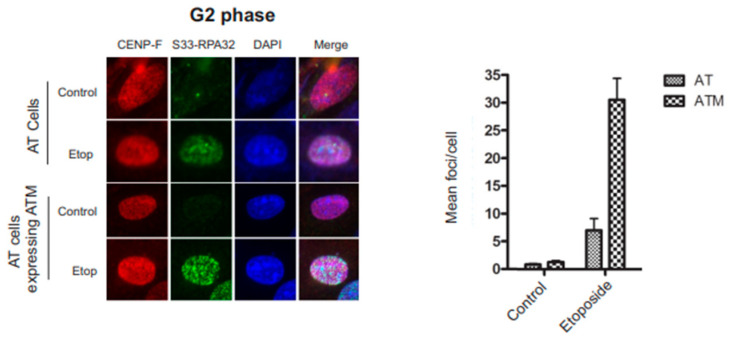
G2 phase synchronized AT cells and AT cells expressing ATM were treated with vehicle or 20 µM etoposide for 2 h. Cells were detergent washed and fixed, and immunofluorescence analysis of RPA32 phosphorylated at Ser33 was performed. G2 cells were identified using CENP-F. A total of 200 cells were assayed for each experiment. The fractions of cells that displayed RPA32 phosphorylated at Ser33 are shown in the right panel. Error bars: SDs from three independent experiments (*n* = 3).

**Figure 5 genes-14-02205-f005:**
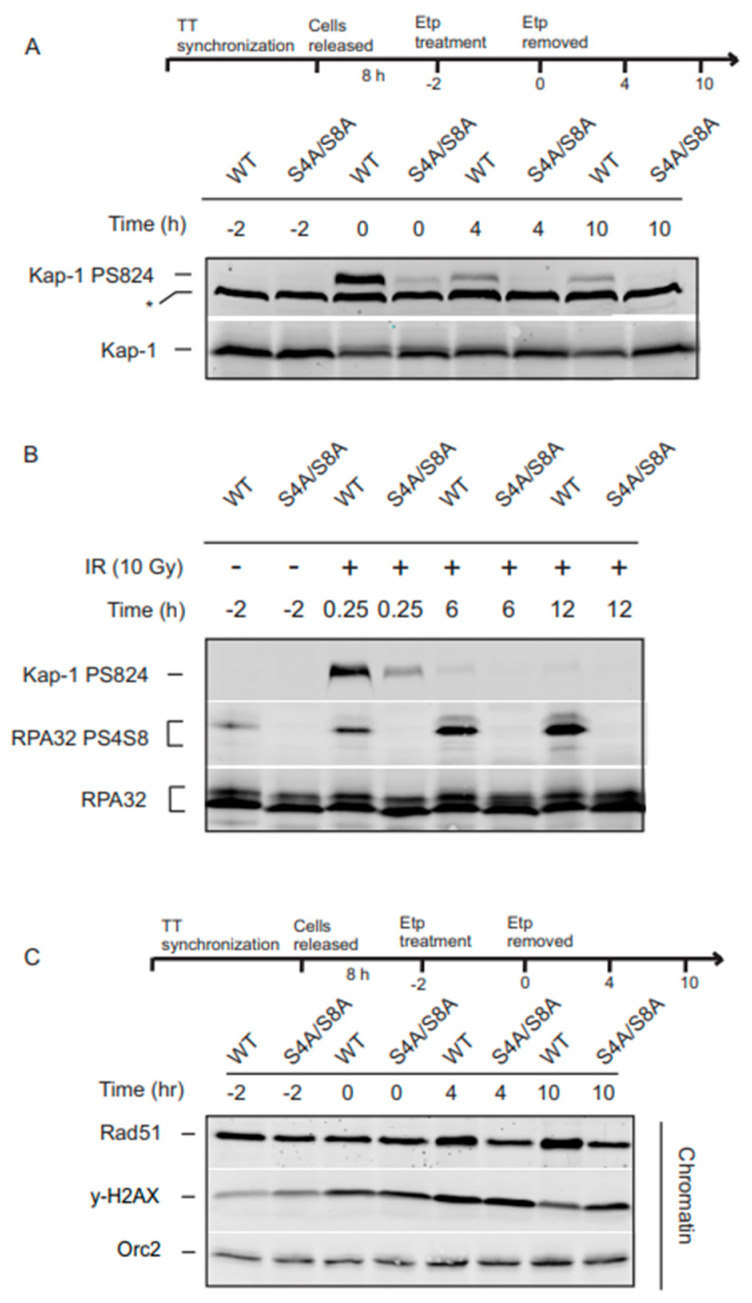
(**A**) Cells were treated with etoposide (20 µM, from −2 to 0 h) and examined for levels of KAP-1 Ser824 phosphorylation. Non-specific band indicated by asterisk (*). (**B**) Cells were analyzed for KAP-1 phosphorylation after treatment with IR (10 Gy). In a and b, cell lysates were prepared, resolved by SDS-PAGE and immunoblotted with antibodies to RPA32 and phosphorylated KAP-1, RPA32 Ser4/Ser8. (**C**) Protein levels of phosphorylated H2AX and Rad51 were examined during (20 µM, from −2 to 0 h) and after etoposide removal (0 h, 4 h, 10 h) in chromatin-fractionated extracts. Orc2 was used as a loading control.

## Data Availability

Data is unavailable due to privacy restrictions.

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
