# Peer review of "Role of RPA Phosphorylation in the ATR-Dependent G2 Cell Cycle Checkpoint"

_genes, 2023, doi:10.3390/genes14122205_

Round 1
Reviewer 1 Report
Comments and Suggestions for Authors
Liu et al., in the present manuscript investigated the role of RPA32 Ser4/Ser8 phosphorylation in response to DNA damage, from S to G2 phase of the cell cycle with regard to TopBP1 and Rad9 accumulation on chromatin and activation of the ATR-dependent G2 checkpoint. The data provided would suggest that RPA Ser4/Ser8 phosphorylation controls ATM-dependent KAP-1 phosphorylation and Rad51 chromatin loading in G2 cells.
The manuscript by Liu et al., is interesting but not novel and lacks of controls.
- Have the authors quantified by other biochemical approaches the induction of DSBs after Etoposide treatment? What is the degree of DNA-DSBs induced compared to controls? Do the authors would expect cyclins inhibitors have similar outcomes?
- It would be useful if the authors in Figure 1 could provide the percentage of the cells in the different phases of the cell cycle during the synchronization process.
Author Response
Thank you for your review and suggestions.
We have previously shown with similar dosing of etoposide that g-H2AX foci which are a marker of DSBs is induced in cells damaged with etoposide compared to controls (Liu S et al. (2012) Nucleic Acids Res 40:10780-10794. While the level of g-H2AX foci is not a direct measurement of DSBs, others have shown at least, at moderate to high levels of etoposide doses similar to that we used, through PFGE and other direct methods of detecting DSBs that g-H2AX is a useful indicator of the relative amount of DSBs.
Lobrich M, Shibata A, Beucher A, Fisher A, Ensminger M, Goodarzi AA, Barton O and Jeggo PA (2010) gammaH2AX foci analysis for monitoring DNA double-strand break repair: strengths, limitations and optimization. Cell Cycle 9:662-9.
Bouquet F, Muller C and Salles B (2006) The loss of gammaH2AX signal is a marker of DNA double strand breaks repair only at low levels of DNA damage. Cell Cycle 5:1116-22.
Do the authors expect cyclins inhibitors to have similar outcomes? Interestingly, it has been shown with a CDK1 inhibitor that this inhibitor induced G2 arrest and prevented cells with DNA double-strand breaks from transitioning into the M-phase. This is the exact opposite of what a lack of RPA32 phosphorylation does to cells.
Sure, a percentage of the cells in the different phases during synchronization can be added.
Reviewer 2 Report
Comments and Suggestions for Authors
The DNA double-strand break lesion is a serious problem for cell repair machinery especially if DSB was formed as a clustered lesion (CL). The DSB CL can lead to genome molecule instability and in consequence, can be lethal for cells. Due to the above the investigation of cell response to this type of damage is important from the therapeutic and toxicological points of view. The author in their article entitled Role of RPA phosphorylation in the ATR-dependent G2 cell cycle Checkpoint the role of two proteins for which the Ser4/Ser8 phosphorylation dephosphorylation and ubiquitylation are crucial. The above poses some implications on the biochemistry of the 9-1-1 complex as well as on TopBP1 which can lead to the activity changes of ATR. I have found the article interesting and valuable for broad scientific audiences. Moreover, the article is well-written and readable however I notice some typing mistakes.
The critical remarks:
1- In the abstract authors should put a sentence about the significance of their discovery
2- The source of DSB should be mentioned in the introduction part as well as the role of BER in their formation (obligatory)
3- The numbers of DSB in pre and post-irradiative cells (tissue)
4- In the experimental part, I did not notice the control which proves that in the investigated system DSBs were present
5- the last part of the introduction is, in fact, the conclusion and should be moved to the end of the article
Author Response
- A sentence of significance has been added to the end of the abstract.
- The source of DNA damage has been added to the introduction.
- The number of DSBs from etoposide has previously been calculated to be 200 DSBs/cell per 1 uM of etoposide. Muslimović A, Nyström S, Gao Y, Hammarsten O. Numerical analysis of etoposide induced DNA breaks. PLoS One. 2009, 4: e5859
- As previously mentioned with Reviewer 1:
We have previously shown with similar dosing of etoposide that g-H2AX foci which are a marker of DSBs is induced in cells damaged with etoposide compared to controls (Liu S et al. (2012) Nucleic Acids Res 40:10780-10794. While the level of g-H2AX foci is not a direct measurement of DSBs, others have shown at least, at moderate to high levels of etoposide doses similar to that we used, through PFGE and other direct methods of detecting DSBs that g-H2AX is a useful indicator of the relative amount of DSBs.
Lobrich M, Shibata A, Beucher A, Fisher A, Ensminger M, Goodarzi AA, Barton O and Jeggo PA (2010) gammaH2AX foci analysis for monitoring DNA double-strand break repair: strengths, limitations and optimization. Cell Cycle 9:662-9.
Bouquet F, Muller C and Salles B (2006) The loss of gammaH2AX signal is a marker of DNA double strand breaks repair only at low levels of DNA damage. Cell Cycle 5:1116-22.
-
We have removed this part of the introduction and added it to the Discussion.
Round 2
Reviewer 1 Report
Comments and Suggestions for Authors
The authors have considered/addressed only part of the concerns previously raised.
Reviewer 2 Report
Comments and Suggestions for Authors
The recent version of the article consists of all the necessary changes. Therefore I can recommend it for publication.